# Intracellular Signaling Responses Induced by Radiation within an In Vitro Bone Metastasis Model after Pre-Treatment with an Estrone Analogue

**DOI:** 10.3390/cells10082105

**Published:** 2021-08-17

**Authors:** Jolene Helena, Anna Joubert, Peace Mabeta, Magdalena Coetzee, Roy Lakier, Anne Mercier

**Affiliations:** 1Department of Physiology, School of Medicine, Faculty of Health Sciences, University of Pretoria, Pretoria 0001, South Africa; jolenehelena@gmail.com (J.H.); annie.joubert@up.ac.za (A.J.); peace.mabeta@up.ac.za (P.M.); magdalena.coetzee55@gmail.com (M.C.); 2Department of Radiation Oncology, Steve Biko Academic Hospital, School of Medicine, Faculty of Health Sciences, University of Pretoria, Pretoria 0001, South Africa; roy.lakier@up.ac.za

**Keywords:** cancer, bone metastasis, angiogenesis, osteoclasts, osteoblasts, ESE-16, radiosensitization, apoptosis

## Abstract

2-Ethyl-3-O-sulfamoyl-estra-1,3,5(10)16-tetraene (ESE-16) is an in silico-designed estradiol analogue which has improved the parent compound’s efficacy in anti-cancer studies. In this proof-of-concept study, the potential radiosensitizing effects of ESE-16 were investigated in an in vitro deconstructed bone metastasis model. Prostate (DU 145) and breast (MDA-MB-231) tumor cells, osteoblastic (MC3T3-E1) and osteoclastic (RAW 264.7) bone cells and human umbilical vein endothelial cells (HUVECs) were representative components of such a lesion. Cells were exposed to a low-dose ESE-16 for 24 hours prior to radiation at non-lethal doses to determine early signaling and molecular responses of this combination treatment. Tartrate-resistant acid phosphatase activity and actin ring formation were investigated in osteoclasts, while cell cycle progression, reactive oxygen species generation and angiogenic protein expression were investigated in HUVECs. Increased cytotoxicity was evident in tumor and endothelial cells while bone cells appeared to be spared. Increased mitotic indices were calculated, and evidence of increased deoxyribonucleic acid damage with retarded repair, together with reduced metastatic signaling was observed in tumor cells. RAW 264.7 macrophages retained their ability to differentiate into osteoclasts. Anti-angiogenic effects were observed in HUVECs, and expression of hypoxia-inducible factor 1-α was decreased. Through preferentially inducing tumor cell death and potentially inhibiting neovascularization whilst preserving bone physiology, this low-dose combination regimen warrants further investigation for its promising therapeutic application in bone metastases management, with the additional potential of limited treatment side effects.

## 1. Introduction

Metastasis is an inefficient process with an estimate of only 0.01% of tumor cells successfully disseminating from a solid tumor and establishing themselves at a distant site [1]. However, bone metastasis is highly prevalent in breast and prostate cancer patients [2]. Establishment of bone metastatic lesions involves a complex interaction between tumor cells, bone-forming and -resorbing cells, angiogenesis, and mineralized bone matrix to form a supportive microenvironment to foster secondary tumor formation [1,3,4].

Metastasis relies on the migratory and invasive properties of propagating tumor cells to form distant lesions. Undifferentiated tumor cells undergo epithelial–mesenchymal transition (EMT) which is brought about by genetic and epigenetic signaling in the tumor microenvironment [5]. Tumor-associated neovasculature is essential for the delivery of nutrients and oxygen to the tumor microenvironment and the removal of metabolic waste and carbon dioxide from the tumor microenvironment. Therefore, tumorigenesis and tumor survival depend on the induction of angiogenesis [6,7,8].

Migratory tumor cells that have lodged in the bone microenvironment secrete factors such as parathyroid hormone-related protein (PTHrP) that stimulate osteoblasts, and have a complex interplay of signaling between the different histological components encompassing growth factors, chemokines, cytokines, proteases, and vascular promoting factors [3,9,10]. Activated osteoblasts trigger receptor activator of nuclear factor-κB ligand (RANKL) expression which binds to the receptor activator of nuclear factor-κB (RANK) on macrophages to mediate differentiation into mature osteoclasts [3,9]. Activated osteoclasts degrade bone matrix with the aid of strong acids and proteinases such as cathepsins and matrix metalloproteinases (MMPs) [3,9]. Chronic and persistent inflammation also plays a strong role in the tumor cell homing and establishment in the bone microenvironment [11]. Targeted intervention is crucial in an attempt to disrupt the pathological relationship between tumor and bone cells within the bone microenvironment whilst preserving bone integrity and homeostasis.

The osteotropic behavior of tumor cells is driven by bone-specific metastatic signaling molecules such as matrix metalloproteinase-9 (MMP-9) and bone morphogenetic protein-7 (BMP-7), which are highly expressed in prostate and breast cancers and promote EMT and bone invasion [12,13]. Osteoblastic bone metastasis involves the stimulation of bone formation, common in prostate cancer patients. Osteolytic bone metastasis are characterized by excessive bone resorption and the inhibition of bone formation, often observed in various types of breast cancer [14]. In osteoblastic bone lesions, the disrupted alignment of collagen fibrils and biological apatite in long bones, together with the formation of low mineral density bone, impairs bone mechanical function [15]. Skeletal complications are common in both osteoblastic and osteolytic bone metastases and include severe pain, pathological fractures, and nerve compression, as well as anemia [14,16].

The eukaryotic cytoskeleton is a complex network composed of intermediate filaments for mechanical support and resistance to shear stress; actin for cell polarity, motility, shape, and cytokinesis; and microtubules for organelle positioning, intracellular trafficking, and chromosome segregation during mitosis [17,18]. Microtubule-targeting agents (MTAs) suppress the microtubule dynamic instability parameters leading to mitotic arrest and cell death [19,20]. Paclitaxel stabilizes the microtubules and has demonstrated radiosensitizing properties in vitro through inducing a G_2_/M phase arrest, the cell cycle phase with the greatest relative radiosensitivity [21,22]. On the other hand, microtubule-destabilizing drugs such as vinblastine and colchicine decrease polymerization and abrogate the cytoskeletal anatomy [23,24]. Analogues of MTAs are common in the clinical treatment of various cancers, and research continues in an effort to improve efficacy, overcome drug resistance, and reduce adverse effects in this class of chemotherapeutics [25,26].

2-Methoxyestradiol (2ME2) is an endogenous 17β-estradiol metabolite with anti-mitogenic and anti-angiogenic properties demonstrated in vitro and in vivo (Figure 1) [27]. 2ME2 exerts anti-tumorigenic effects by interacting with the colchicine binding sites on tubulin, thus limiting microtubule polymerization and disrupting mitotic spindle dynamics in a dose- and time-dependent manner [28]. Casarez et al. demonstrated the radiosensitizing effects of 2ME2 in in vitro and in vivo prostate carcinoma models [29]. Amorino et al. proposed that the mechanism of radiosensitization by 2ME2 entailed the redistribution and accumulation of cells in the G_2_/M phase resulting in a repair-deficient cell population that is thus radiosensitive [30]. Zhao et al. observed enhanced radiosensitivity in the radioresistant human melanoma MDA-MB-435R cell line when pre-treated with 2ME2 through hypoxia-inducible factor 1-α (HIF-1α) inhibition [31]. Eriksson et al. indicated bone-sparing effects of 2ME2 in orchidectomized mice through a mechanism involving the reversed conversion of 2ME2 to 17β-estradiol [32]. Advantages of 2ME2 include tolerable side effects, preferential sparing of non-tumorigenic tissue and retained cytotoxicity in multi-drug resistant cancers [33,34]. A major shortcoming of 2ME2 is its rapid hepatic metabolism by 17β-hydroxysteroid dehydrogenase type 2, resulting in low bioavailability [33,35].

Stander et al. designed a range of 2ME2 analogues in silico in an attempt to overcome the pharmacokinetic constraints of the parent compound by improving its bioavailability, potency, and tumor localization due to preferential carbonic anhydrase IX binding in the acidic micromilieu [33,36,37]. 2-Ethyl-3-*O*-sulfamoyl-estra-1,3,5(10)16-tetraene (ESE-16) is one of these sulfamoylated 2ME2 analogues that is not commercially available (Figure 1) [38,39]. ESE-16 is not a substrate for P-glycoprotein pumps, acts independently of hormone receptors, and demonstrates unchanged oral absorption in murine models [33,40]. ESE-16 has demonstrated anti-tumorigenic effects at nanomolar concentrations in vitro through a crosstalk mechanism involving apoptosis and autophagy [39,41]. The estrone analogue disrupts microtubule dynamics and causes metaphase arrest in various cancer cell lines, including oestrogen receptor-positive MCF-7 and oestrogen receptor-negative MDA-MB-231 breast cancer cells, whilst sparing non-tumorigenic cells as well as erythrocytes, fibrin networks, and platelets [38,41,42,43].

Radiation is frequently used in the treatment of bone metastases. However, it is accompanied by general and skeletal adverse effects. Using MTAs as radiosensitizers may augment nuclear damage and delay the repair thereof by impairing nucleocytoplasmic shuttling of DNA damage and repair signaling proteins [44,45]. The effects of the ESE-16 in combination with radiation has not been investigated in tumorigenic prostate cells and non-tumorigenic cell lines such as those of the bone and endothelium. The aim of this study was to investigate the cellular responses of breast and prostate cancer bone metastases receiving radiation following pre-sensitization with ESE-16 using a deconstructed in vitro model comprised of tumor, bone, and endothelial cells. This proof-of-concept study investigated the individual responses of the representative components of bone metastasis and provides insight into the independent intracellular signaling pathways as well as the potential modification of the ECM.

## 2. Materials and Methods

### 2.1. Cell Lines

Metastatic human prostate carcinoma cells (DU 145) and breast adenocarcinoma cells (MDA-MB-231); murine pre-osteoblasts (MC3T3-E1) and macrophages (RAW 264.7); and human umbilical vein endothelial cells (HUVECs) were used in this in vitro study. All cell lines are commercially available and were purchased from American Type Culture Collection (ATCC) (Maryland (MD), United States of America (USA)).

### 2.2. General Cell Culture and Reagents

DU145, MDA-MB-231, MC3T3-E1, and RAW 264.7 cells were cultured in GibcoTM Dulbecco’s modified Eagle medium (DMEM) (Thermo Fisher Scientific, Waltham, Massachusetts (MA), USA) supplemented with penicillin G (100 U/mL), fungizone (250 µg/L), streptomycin (100 µg/L) (Sigma–Aldrich, St Louis, Missouri (MO), USA), and 10% heat-inactivated HyClone^TM^ fetal bovine serum (FBS) (Thermo Fisher Scientific, Waltham, MA, USA). HUVECs were cultured in Clonetics^TM^ endothelial basal medium (EBM)^TM^ supplemented with 2% FBS, penicillin G (10 μg/mL), amphotericin B (25 ng/mL), gentamicin (10 μg/mL), heparin sulfate (0.75 μg/mL), hydrocortisone (1 μg/mL), ascorbic acid (50 μg/mL), insulin growth factor (IGF) (15 ng/mL), vascular endothelial growth factor-A (VEGF-A) (5 ng/mL), epidermal growth factor (EGF) (5 ng/mL), basic fibroblast growth factor (bFGF) (5 ng/mL), and L-glutamine (10 mM) (Lonza, Basel, Switzerland). Cell culture was conducted in sterile conditions using a Biobase^®^ laminar flow cabinet (Jinan, China) and cells were incubated in a Forma Scientific water-jacketed incubator (37 °C, 5% CO_2_) (Thermo Fisher Scientific, Waltham, MA, USA). Cell culture growth medium was replaced every 3 days. Cells were washed with 1× phosphate buffer saline (PBS) (10×: 80 g/L NaCl, 11.5 g/L Na_2_HPO_4_, 2 g/L KCl, 2 g/L KH_2_PO_4_). RAW 264.7 macrophages were scraped, and DU 145-, MDA-MB-231-, MC3T3-E1 cells and HUVECs were trypsinized with trypsin-ethylenediaminetetraacetic acid (EDTA) (Thermo Fisher Scientific, Waltham, MA, USA). All chemicals not specifically stated were of analytical grade and were purchased from Sigma–Aldrich (St Louis, MO, USA).

### 2.3. Compound and Experimental Protocols

ESE-16 is not commercially available and was synthesized by iThemba Pharmaceuticals (Pty) Ltd. (Johannesburg, Gauteng, South Africa (SA)). ESE-16 was dissolved in dimethyl sulfoxide (DMSO) (Sigma–Aldrich, St Louis, MA, USA) to a 10-mM stock solution and stored at −20 °C. Exponentially growing cells were seeded at appropriate densities for each experiment, with a 24-h cell attachment policy. The lowest ESE-16 concentration to significantly induce apoptosis and the lowest radiation dose to significantly reduce cell viability were determined in the experimental dose–response curve set-up (Supporting data). DU 145 cells, RAW 264.7 macrophages, and HUVECs were exposed to 0.235 µM ESE-16 and MDA-MB-231 cells were exposed to 0.176 µM of ESE 16 for 24 h. After the 24-h ESE-16-exposure, cells were exposed to a single dose of 4 Gy X-radiation. Termination proceeded 2, 24, and 48 h after irradiation.

### 2.4. Radiation

Samples were irradiated using a Siemens Oncor Impression Linear Accelerator (Siemens Medical Solutions, Malvern, Pennsylvania (PA), USA) at the Department of Radiation Oncology, Steve Biko Academic Hospital (Pretoria, Gauteng, SA). Photon beam energy of 6 MV was used with a direct field and a gantry angle of 180°. Source-to-surface distance (SSD) was 100 cm and the photon beam electronic equilibrium depth (Dmax) was 1.5 cm. A field size of 10 × 10 cm was used with 4.5-cm tissue equivalent bolus for dose homogeneity.

### 2.5. Controls

Cells propagated in DMEM served as a negative control and DMSO (0.05% *v/v*) served as a vehicle control. No statistical differences were observed between the negative and vehicle controls. The ESE-16 treatment control entailed ESE-16 exposure at 24 h after seeding for the experimental duration. The radiation treatment control followed the same experimental timeline with irradiation at 48 h after seeding. Paclitaxel served as a positive method control for apoptosis, 2ME2 for mitotic arrest, and etoposide for DNA damage.

### 2.6. Cytotoxicity Studies: Dose–Response Curves of the Compound with and without Radiation on All Cell Lines

Exponentially growing cells were exposed to a dilution series of ESE-16 for 48 h (10, 5, 2.5, 1.25, 0.625, 0.313, 0.156, 0.078, 0.039, and 0.02 µM). In parallel, cells were exposed to the same dilution series and received 4 Gy radiation after 24 h of ESE-16 exposure. Sodium dodecyl sulfate (SDS) (0.1%) served as the positive control (0% cell viability) and DMSO (0.05%) served as the negative control (100% cell viability). Cells were washed with PBS (37 °C) and 0.5 mg/mL 3-(4,5-dimethylthiazol-2-yl)-2,5-diphenyltetrazolium bromide (MTT) (Sigma–Aldrich, St Louis, MO, USA) in Roswell Park Memorial Institute (RPMI) medium (Sigma–Aldrich, St Louis, MO, USA) (37 °C) was added (37 °C, 4-h). The solubilization solution (isopropanol, 0.1 N HCl, 10% SDS, 10% TritonTM X-100) was added (room temperature (RT)) overnight, protected from light). Absorbance was measured at 570 nm using an ELx800 Absorbance Reader (BioTek Instruments, Inc., Winooski, VT, USA) and half maximal growth inhibition concentration (GI50) values were calculated.

### 2.7. Cell Cycle Progression Assessment Using Flow Cytometry

Trypsinized cells were washed with ice-cold PBS containing 0.1% FBS and fixed with ice-cold 70% ethanol (4 °C, overnight). The next day, cells were resuspended in PBS containing 0.01% Triton^TM^ X-100, 40 μg/mL propidium iodide (PI) and 100 μg/mL ribonuclease A (RNase A) (Sigma–Aldrich, St Louis, MO, USA) (37 °C, 40 min, protected from light). PI (FL3) fluorescence was measured on a FC 500 series flow cytometer (Beckman Coulter, Brea, CA, USA).

### 2.8. Cell Division: Mitotic Index Determination Using Light Microscopy

Cells were fixed with Bouin’s solution (Sigma–Aldrich, St Louis, MO, USA) (RT, 30 min), followed by washing with 70% ethanol (20 min) and double-distilled water (ddH_2_O). Cells were stained with Harris modified hematoxylin solution (Sigma–Aldrich, St Louis, MO, USA) (RT, 20 min), followed by washing with ddH_2_O and 70% ethanol. Cells were stained with 1% eosin (British Drug Houses, Ltd., London, UK) in 70% ethanol (RT, 7 min). Cells were washed twice (5 min each) with ethanol (70%, 96%, 100%) and xylol. Entellan^®^ mounting fluid (Merck, Darmstadt, Germany) was used to mount coverslips onto slides. A Nikon Optiphot transmitted light microscope (Tokyo, Japan) was used for visualization. The cells were scored blind, with slides deidentified, and were only linked to each treatment condition after the analysis was concluded.

### 2.9. Apoptosis: Annexin V-FITC Detection by Flow Cytometry

The Annexin V-fluorescein isothiocyanate (FITC) Apoptosis Detection Kit (BioLegend, Inc., San Diego, CA, USA) was used. Trypsinized cells were resuspended in 100 µL of Annexin V binding buffer, to which 5 μL of Annexin V-FITC and 10 μL of PI were added (RT, 15 min, protected from light). An additional 400 µL of Annexin V binding buffer was added. FITC (FL1) and PI (FL3) fluorescence were measured on a FC 500 series flow cytometer (Beckman Coulter, Brea, CA, USA).

### 2.10. Apoptotic Signaling: Colorimetric Determination of Caspase 3 Activity

The Caspase 3/cysteine protease P32 (CPP32) Colorimetric Assay Kit (BioVision, Inc., Milpitas, CA, USA) was used. Trypsinized cells were resuspended in ice-cold cell lysis buffer (stored on ice, 10 min). Cytosolic extract was obtained by centrifugation (10,000× *g*, 10 min, 4 °C). Protein determination was performed using the Pierce^TM^ bicinchoninic acid (BCA) Protein Assay Kit (Thermo Fisher Scientific, Waltham, MA, USA). Protein (100 µg) was diluted in cell lysis buffer, to which an equal volume of 2× reaction buffer containing 10 mM dithiothreitol (DTT) was added. Aspartate-glutamate-valine-aspartate (DEVD)-*p*-nitroaniline (*p*NA) (4 mM) was subsequently added (37 °C, 2 h, protected from light). Absorbance was measured at 405 nm using an ELx800 Absorbance Reader (BioTek Instruments, Inc., Winooski, VT, USA).

### 2.11. Reactive Oxygen Species: Superoxide Detection by Flow Cytometry

Trypsinized cells were resuspended in 1 µM hydroethidine (Sigma–Aldrich, St Louis, MO, USA) stock solution in PBS (37 °C, 20 min, protected from light). Hydroethidine (FL3) fluorescence was measured on a Gallios flow cytometer (Beckman Coulter, Brea, CA, USA).

### 2.12. DNA Damage: Micronuclei Quantification Using Light Microscopy

Samples were treated with 5.6 µg/mL cytochalasin-B (Sigma–Aldrich, St Louis, MO, USA) for the duration of one complete cell cycle. Trypsinized cells were washed with PBS and pelleted (90× *g*, 15 min, 4 °C). Hypotonic solution (0.14 M KCl; 37 °C) was added (RT, 5 min) and cells were pelleted (90× *g*, 15 min, 4 °C). Fixative 1 (13 parts 0.9% NaCl, 12 parts methanol, 3 parts acetic acid) was added (RT, 5 min) and cells pelleted (100× *g*, 10 min, 4 °C). Fixative 2 (4-parts methanol, 1-part acetic acid) was added (RT, 5 min) and cells were pelleted (100× *g*, 10 min, 4 °C). Cells were washed with fixative 2, applied to slides, and dried overnight. Cells were stained with Giemsa (RT, 20 min). A Nikon Optiphot transmitted light microscope (Tokyo, Japan) was used for visualization.

### 2.13. DNA Damage: Flow Cytometric Quantification of Histone H2A.X Phosphorylation

The FlowCellect^TM^ DNA Damage Histone H2A.X Dual Detection Kit (Merck Millipore, Billerica, MA, USA) was used. Trypsinized cells were resuspended in equal parts of 1× wash buffer and 1× fixation buffer (on ice, 10 min), followed by washing with 1× assay buffer. Cells were permeabilized with ice-cold 1× permeabilization buffer (on ice, 20 min), followed by washing with 1× assay buffer. Cells were incubated in 90 μL of 1× assay buffer, 5 μL of anti-histone H2A.X-FITC, and 5 μL of anti-phospho-histone H2A.X (Ser139)-peridinin chlorophyll protein (PerCP) (RT, 30 min, protected from light), then washed twice with 1× assay buffer. Anti-histone H2A.X-FITC (FL1) and anti-phospho-histone H2A.X (Ser139)-PerCP (FL4) fluorescence were measured on a Gallios flow cytometer (Beckman Coulter, Brea, CA, USA).

### 2.14. DNA Damage: γ-H2A.X Foci Using Fluorescence Microscopy

Cells were fixed with 2% paraformaldehyde (RT, 20 min), permeabilized with 0.2% Triton^TM^ X-100 (RT, 5 min), and incubated in 2% bovine serum albumin (BSA) in 0.2% PBS-TWEEN^®^ 20 (RT, 1-h). Cells were incubated in 1:500 primary mouse anti-γ-H2A.X (phospho Ser139) antibody (Abcam, Cambridge, England; ab26350) (RT, 30 min), and washed. Cells were subsequently incubated in 1:1000 anti-mouse Alexa Fluor^®^ 488-labeled secondary antibody raised in donkey (Invitrogen, CA, USA; A-21202) and 1:1000 4’,6-diamidino-2-phenylindole (DAPI) (Sigma–Aldrich, St Louis, MO, USA) (RT, 30 min, protected from light). Washing was performed thrice and coverslips were mounted onto slides using Fluoromount^TM^ aqueous mounting medium (stored at 4 °C). A Zeiss LSM 880 confocal laser scanning microscope with Airyscan (Jena, Germany) at the Laboratory for Microscopy and Microanalysis at the University of Pretoria (Pretoria, South Africa) was used to visualize Alexa Fluor^®^ 488 (green fluorescence) excited at 490 nm and emitted at 525 nm, and DAPI (blue fluorescence) excited at 358 nm and emitted at 461 nm.

### 2.15. Metastatic Signaling: Western Blot Analyses of BMP-7 and MMP-9 Expression

Cells were lysed with ice-cold radioimmunoprecipitation assay (RIPA) buffer (150 mM of NaCl, 10 mM of Tris-HCl, pH 7.4, 0.1% SDS, 0.5% sodium deoxycholate, 1 mM of EDTA, 1 mM of ethylene glycol tetraacetic acid (EGTA), 1:100 protease inhibitor, and phosphatase inhibitor cocktail 2 (Sigma–Aldrich, St Louis, MO, USA)) (stored on ice, 5 min). Lysed cells were scraped and cytosolic extract was obtained by centrifugation (10,000× *g*, 30 min, 4 °C). Protein determination was performed using the Pierce^TM^ BCA Protein Assay Kit (Thermo Fisher Scientific, Waltham, MA, USA), as previously described. Proteins were denatured with 4× NuPAGE^TM^ lithium dodecyl sulfate (LDS) buffer (Thermo Fisher Scientific, Waltham, MA, USA) containing 2.5% 2-mercaptoethanol (95 °C, 5 min). Protein (25 μg) was loaded onto NuPAGE^TM^ 4–12% Bis-Tris protein gels (Thermo Fisher Scientific, Waltham, MA, USA). Gel electrophoresis was performed in 1× 3-morpholinopropane-1-sulfonic acid (MOPS) buffer (20×: 50 mM MOPS, 50 mM Tris, 1 mM EDTA, 0.1% SDS, pH 7.7) against SeeBlue^®^ Plus2 Pre-Stained Protein Standard (Thermo Fisher Scientific, Waltham, MA, USA) (RT, 90 V, 3-h). Proteins were transferred onto Immun-Blot^®^ polyvinylidene fluoride (PVDF) membranes (Bio-Rad Laboratories, Inc., Hercules, CA, USA) (activated in 100% methanol) using 1× transfer buffer (10×: 25 mM Tris, 192 mM glycine, pH 8.3) (4 °C, 100 V, 1.5-h). Membranes were incubated in 2.5% BSA in 0.2% PBS-TWEEN^®^ 20 (RT, 1 h, agitated), followed by incubation in 1 µg/mL primary mouse anti-BMP-7 antibody (Abcam, Cambridge, UK; ab54904) or 1:1000 primary rabbit anti-MMP-9 antibody (Abcam, Cambridge, UK; ab38898) (4 °C, overnight, agitated). Membranes were incubated in 1:10,000 anti-mouse (Abcam, Cambridge, UK; ab97023) or anti-rabbit (Abcam, Cambridge, UK; ab97051) horse radish peroxidase (HRP)-labeled secondary antibody raised in goat (RT, 1 h, agitated). Chemiluminescence was detected using Pierce^TM^ enhanced chemiluminescence (ECL) Western blotting substrate on a ChemiDoc^TM^ MP Imaging System (Bio-Rad Laboratories, Inc., Hercules, CA, USA). Proteins were standardized with 1:5000 mouse anti-β-actin antibody (Sigma–Aldrich, St Louis, MO, USA; ab134032) (RT, 1 h, agitated) and 1:10,000 anti-mouse HRP-labeled secondary antibody (Abcam, Cambridge, UK; ab97023) (RT, 1 h, agitated). Species-specific secondary antibodies labeled with horseradish peroxidase were used to obtain a signal, and densitometry analysis was done for semi-quantification using ImageJ software (NIH). Blots were adjusted against loading controls, and expressed as fold-decrease or -increase compared to DMSO vehicle-control cells.

### 2.16. Angiogenic Signaling: Western Blot Analysis of HIF-1α Expression

Western blotting was performed as described above. HIF-1α expression was determined using 1:1000 primary rabbit anti-HIF-1α antibody (Novus Biologicals, Littleton, CO, USA; NB100-479) with 1:10,000 anti-rabbit HRP-labeled secondary antibody (Abcam, Cambridge, UK; ab97051).

### 2.17. Effect of the Combination Treatment on the Bone Component of the Metastatic Lesion

For TRAP activity and staining, RAW 264.7 macrophages were exposed to ESE-16 for 24 h prior to 4 Gy irradiation then differentiated over 5 days in the presence of 15 ng/µL RANKL. In parallel, RAW 264.7 macrophages were differentiated over 5 days in the presence of 15 ng/µL RANKL then treated with ESE-16 for 24 h followed by 4 Gy irradiation. For actin ring formation, RAW 264.7 macrophages were fully differentiated over 5 days in the presence of 30 ng/µL RANKL as previously described, then treated with ESE-16 for 24 h followed by 4 Gy irradiation [46]. Growth medium was replaced after 3 days at which another supplement of RANKL was added.

### 2.18. Osteoclast Differentiation: TRAP Activity and Staining

For intracellular TRAP activity, TRAP reaction buffer (80 µL) (12.5% 0.009 g/mL of L-ascorbic acid, 12.5% 0.046 g/mL of disodium tartrate dihydrate, 12.5% 0.025 g/mL of 4-nitrophenylphosphate disodium hexahydrate, 25% reaction buffer (0.5% Triton^TM^ X-100, 1 M of acetate, 1 M of NaCl, 10 mM of EDTA, pH 5.5), 37.5% ddH_2_O) was added to 20 µL of conditioned medium (37 °C, 5 min, protected from light). The reaction was terminated with stop solution (0.3 M NaOH). Absorbance was measured at 405 nm using an ELx800 Absorbance Reader (BioTek Instruments, Inc., Winooski, VT, USA).

Intracellular TRAP staining was performed by fixing cells with 3.7% formaldehyde (RT, 5 min). Acetate-tartrate buffer (0.1 M of sodium tartrate, 0.2 M of acetate buffer, pH 5.2) was added (RT, 5 min). Solution A (20 mg/mL of naphthol AS-BI phosphate/dimethylformamide in acetate-tartrate buffer) was added (37 °C, 30 min), followed by solution B (acetate-tartrate buffer hexazotized pararosaniline solution) (37 °C, 15 min). Cells were subsequently stained with hematoxylin (RT, 1 min) and dried overnight. A Nikon Optiphot transmitted light microscope (Tokyo, Japan) was used for visualization. TRAP-positive multinucleated cells containing more than three nuclei were counted as osteoclasts.

### 2.19. Osteoclast Activity: Actin Ring Formation Using Fluorescence Microscopy

Treated cells were fixed with 3.7% formaldehyde (RT, 5 min) and permeabilized with 0.2% Triton^TM^ X-100 (RT, 10 min). Cells were incubated in 50 µg/mL of Phalloidin–Atto 488 (Sigma–Aldrich, St Louis, MO, USA) (37 °C, 40 min, protected from light) and 1 µg/µL of Hoechst 33342 (Sigma–Aldrich, St Louis, MO, USA) (37 °C, 5 min, protected from light). A Zeiss AxioCam MRc5 camera attached to a Zeiss Axiovert 40 CFL microscope (Oberkochen, Germany) was used to visualize Phalloidin–Atto 488-stained actin rings (green fluorescence) excited at 488 nm and emitted at 590 nm, and Hoechst 33342-stained nuclei (blue fluorescence) excited at 350 nm and emitted at 461 nm.

### 2.20. Statistical Analyses

Qualitative data were obtained from confocal and fluorescence microscopic techniques performed in triplicate. Semi-quantitative (mitotic index (MI), micronuclei (Mn) and TRAP staining), and quantitative data (spectrophotometry, flow cytometry, and Western blotting) were obtained from a minimum of three independent biological repeats (n ≥ 3) in the cytotoxicity studies. Flow cytometric data from a minimum of 10,000 cells was analyzed using Kaluza Analysis Software, version 2.0 (Beckman Coulter, Brea, CA, USA). Western blot analysis was performed on Image Lab, version 6.0 (Bio-Rad Laboratories, Inc., Hercules, CA, USA). Statistical analyses employed the analysis of variance (ANOVA)-single factor model and two-tailed Student’s *t*-test. *p-*values < 0.05 were considered as statistically significant.

## 3. Results

### 3.1. ESE-16 Is Cytotoxic to Cancer, Pre-Osteoclastic, and Endothelial Cells with Enhanced Sensitivity When Combined with Radiation Whilst Pre-Osteoblasts Are Spared

The GI_50_ values of a 48-h ESE-16-exposure was determined spectrophotometrically with a dose–response curve in all cell lines (Table 1 and Figure 2). DU 145 and MDA-MB-231 cells yielded GI_50_ values of 0.625 ± 0.039 µM and 0.469 ± 0.078 µM, respectively. RAW 264.7 macrophages yielded a GI_50_ value of 0.625 ± 0.156 µM with increased sensitivity to 4 Gy radiation (GI_50_ < 0.039 µM). HUVECS yielded a GI_50_ value of 0.156 ± 0.02 µM with increased sensitivity to radiation as observed in RAW 264.7 macrophages. GI_50_ values of more than 10 µM were reported in MC3T3-E1 cells with and without 4 Gy radiation (GI_50_ > 10 µM).

### 3.2. Subsequent Experimental Conditions

ESE-16 and radiation dose–response curves conducted as part of the experimental set-up determined the lowest ESE-16 concentration and radiation dose to induce apoptosis and significantly reduce cell viability (Appendix A). ESE-16 concentrations of 0.235 and 0.176 µM were determined in DU 145 and MDA-MB-231 cells, respectively. A radiation dose of 4 Gy was determined in both cell lines. Thus, seeded attached cells were exposed to their respective dose of the compound for 24 h, after which 4 Gy radiation was administered. Termination followed 2, 24, or 48 h thereafter, depending on the experiment. HUVECs were exposed to 0.235 µM ESE-16 for 24 h, followed by 4 Gy radiation.

### 3.3. ESE-16-Pre-Sensitization Inhibits Cell Cycle Progression by Inducing G_2_/M Phase Arrest and Promoting Sub-G_1_ Phase Accumulation in Cancer and Endothelial Cells

Cell cycle analysis was performed by flow cytometric quantification of PI. Significant increases in the sub-G_1_ population were found in DU 145 cells exposed to ESE-16 alone (34.43 ± 1.82%) and the combination treatment (42.33 ± 7.79%) when compared to DMSO (55.30 ± 2.80%) (Figure 3A). Moreover, the combination treatment yielded a significantly higher sub-G_1_ population than 4 Gy radiation alone (12.08 ± 2.25%). The G_2_/M population was significantly higher in the combination treatment (29.45 ± 2.31%) when compared to DMSO (24.56 ± 1.48%) and ESE-16 in isolation (20.29 ± 0.20%). Similar trends were observed in MDA-MB-231 cells. In HUVECs, the DMSO vehicle control indicated a G_1_ population of 72.78 ± 1.18% with similar treatment effects as the cancer cells (Figure 3A). The sub-G_1_ and G_2_/M populations were significantly higher in the ESE-16 treatment (9.54 ± 2.24% and 17.45 ± 1.06%, respectively) and the combined therapy (11.39 ± 2.37% and 18.65 ± 1.19%, respectively) when compared to DMSO.

Hematoxylin stained negatively-charged nuclear material (purple) and eosin stained positively-charged cytoplasmic contents (pink) (Figure 3B) [47]. Interphase and the different phases of mitosis (prophase, metaphase, anaphase and telophase) were visualized using light microscopy. The MI was calculated by dividing the total number of cells undergoing mitosis by the total number of cells in the population (Figure 3C). DU 145 cells exposed to DMSO yielded a MI of 11.03 ± 1.42%, similar to MDA-MB-231 cells (12.73 ± 1.23%). DU 145 cells exposed to ESE-16 or 4 Gy radiation showed no statistical differences in MI, whereas MDA-MB-231 cells exposed to ESE-16 (15.00 ± 0.52%) had a significantly higher MI than DMSO. ESE-16/4 Gy radiation combined yielded significantly higher MI values in MDA-MB-231 cells (17.90 ± 2.52%), but this was not so in DU 145 cells (20.70 ± 6.84%).

The distribution of cells within the mitotic population was investigated (Figure 3D). DMSO-exposed DU 145 cells showed 35.66 ± 3.79% of cells in prophase, 36.69 ± 2.33% of cells in metaphase, 12.17 ± 3.91% of cells in anaphase, and 7.48 ± 2.84% of cells in telophase. DU 145 cells exposed to the combination treatment demonstrated a significant decrease in prophase cells (27.89 ± 0.75%) and a significant increase in metaphase cells (65.86 ± 1.46%) when compared to DMSO as well as cells exposed to 4 Gy radiation. Anaphase cells (4.71 ± 1.33%) and telophase cells (1.55 ± 1.10%) were also significantly lower in the combination treatment. DMSO-exposed MDA-MB-231 cells showed 46.92 ± 3.07% of cells in prophase, 37.93 ± 5.47% of cells in metaphase, 9.16 ± 4.58% of cells in anaphase, and 5.99 ± 0.75% of cells in telophase. Similar to combination-treated DU 145 cells, MDA-MB-231 cells demonstrated a significant decrease in prophase cells (29.16 ± 3.50%) with a significant increase in metaphase cells (63.80 ± 3.25%) when compared to DMSO as well as cells exposed to 4 Gy radiation. The combination treatment also revealed a significant reduction in telophase cells (1.67 ± 0.54%).

### 3.4. Apoptosis Was Induced in Combination-Treated Cancer Cells, with the Invovlement of CPP32 in MDA-MB-231 Cells

PS is present on the cell’s plasma membrane surface and is externalized with the onset of apoptosis. Annexin V conjugated to FITC binds to PS sites on the cell’s surface [48]. DU 145 cells exposed to DMSO revealed a viable cell population of 87.82 ± 1.18% (Figure 4A). The viable cell population was significantly reduced in all treatment conditions when compared to DMSO: 74.09 ± 1.36% in ESE-16-esposed cells, 82.44 ± 1.33% in 4 Gy-irradiated cells, and 66.96 ± 0.25% in the combined therapy which was further significantly lower than the individual treatments. Significantly higher apoptotic populations were observed in all treatment conditions when compared to DMSO: 24.55 ± 1.19% in ESE-16-exposed cells, 15.97 ± 1.01% in 4 Gy-irradiated cells, and 31.19 ± 1.61% in the combination treatment which was significantly higher than either treatment alone. Similar observations were found in MDA-MB-231 cells (Figure 4A).

CPP32 (Caspase 3) activity was measured spectrophotometrically to determine whether ESE-16 exposure prior to 4 Gy irradiation involves this protein in the execution of apoptosis [49]. At 24 h post-radiation, significant reductions in CPP32 activity were observed in DU 145 cells exposed to ESE-16 alone (0.93 ± 0.04-fold) and the combined therapy (0.92 ± 0.02-fold). In contrast, MDA-MB-231 cells showed significant increases in CPP32 activity in all treatment conditions: ESE-16 (1.11 ± 0.07-fold), 4 Gy radiation (1.16 ± 0.07-fold), and the combination treatment (1.16 ± 0.09-fold) (Figure 4B). At 48 h post-radiation, the combined therapy showed no statistical fold-change (FC) in CPP32 activity in DU 145 cells (1.11 ± 0.11-fold), but significantly higher activity in MDA-MB-23 cells (1.10 ± 0.07-fold).

### 3.5. HUVECs Generate Superoxide Ions in Response to the Combination Treatment

Superoxide (O_2_^−^) was quantified using flow cytometry in order to evaluate reactive oxygen species (ROS) generation in HUVECs at 24 h post-treatment. Overlay histograms representing O_2_^−^ detection revealed significantly higher geometric means in ESE-16-exposed HUVECs (1.19 ± 0.12-fold) as well as HUVECs exposed to ESE-16 in combination with 4 Gy radiation (1.40 ± 0.27-fold) (Figure 4C).

### 3.6. Drug Pretreated Cancer Cells Demonstrated Greater Radiation-Induced DNA Damage

Mn are nuclear fragments indicative of double-strand DNA breaks [50]. The average number of Mn per cell as well as the total number of Mn per treatment condition were documented at 2 and 24 h post-radiation in order to determine the extent of DNA damage. DMSO-exposed DU 145 cells showed that the majority of cells had no Mn, 87.00 ± 6.68% and 84.00 ± 3.29% at 2 and 24 h, respectively. This was also observed in MDA-MB-231 cells, 84.67 ± 2.44% and 84.53 ± 2.93% at 2 and 24 h, respectively (Figure 5A). DU 145 cells exposed to ESE-16 and 4 Gy radiation showed significant decreases in cells without Mn at 2 (39.40 ± 3.67%) and 24 h (37.40 ± 6.66%) when compared to DMSO and ESE-16 alone. Similar observations were found in MDA-MB-231 cells with 32.60 ± 7.40% at 2 h and 37.47 ± 7.49% at 24 h. At 2 h, significant increases in DU 145 cells containing 1 (32.00 ± 2.95%), 2 (20.27 ± 2.61%), 3 (6.53 ± 1.92%), and 4 Mn (1.27 ± 0.31%) were observed in the combination treatment. At the same time point, MDA-MB-231 cells containing 1 Mn (29.07 ± 6.93%), 2 Mn (27.73 ± 4.39%), 3 Mn (8.07 ± 2.10%), and 4 Mn (1.87 ± 0.76%) were also all significantly higher in the combination treatment. Significant increases were maintained in both cell lines at 24 h. These values were significantly higher than in the ESE-16 controls in both cell lines at both time periods. DU 145 cells containing 3 and 4 Mn at 2 h and 1 Mn at 24 h were significantly higher than cells exposed to 4 Gy radiation. The DMSO vehicle control yielded totals of 87.67 ± 46.69 and 107.33 ± 20.88 Mn at 2 and 24 h, respectively, in DU 145 cells, and 104.00 ± 12.49 and 106.00 ± 24.64 Mn, respectively, in MDA-MB-231 cells (Figure 5A). At 2 h, 499.67 ± 60.18 Mn were recorded in combination-treated DU 145 cells and 598.33 ± 15.95 Mn in MDA-MB-231 cells. At 24 h, 408.67 ± 81.71 Mn were recorded in combination-treated DU 145 cells and 565.67 ± 96.40 Mn in MDA-MB-231 cells.

The combination treatment yielded significantly higher total Mn numbers than in the ESE-16 experimental control in both cell lines at both time periods. Radiation is primarily responsible for inducing Mn formation. However, the extent of DNA damage is exacerbated with the addition of ESE-16, as more Mn per cell were evident even though total Mn numbers were similar.

Phosphorylated histone H2A.X (γ-H2A.X) serves as a sensitive biomarker of DNA damage and the extent thereof. H2A.X and γ-H2A.X levels were quantified using flow cytometry [51]. Overlay histograms representing γ-H2A.X expression revealed significantly higher geometric means in combination-treated DU 145- (1.39 ± 0.20-fold) and MDA-MB-231 cells (1.33 ± 0.17-fold) (Figure 5B). Additionally, combination-treated MDA-MB-231 cells revealed significantly higher γ-H2A.X expression than ESE-16 in isolation (0.86 ± 0.14-fold). In isolation, ESE-16 and radiation did not demonstrate significant differences in γ-H2A.X levels when compared to DMSO.

Fluorescent microscopy was employed to detect γ-H2A.X foci within nuclei. Microscopic results correlated with flow cytometric results in both DU 145 and MDA-MB-231 cells (Figure 5B). Treatment with ESE-16 and 4 Gy radiation showed comparable results to the etoposide positive control. Increased γ-H2A.X foci within and outside of nuclei were evident and were accompanied by morphological changes in nuclei.

### 3.7. ESE-16/Radiation Inhibits Metastatic Signaling in Cancer Cells through the Downregulation of BMP-7 and MMP-9 Expression

BMP-7 and MMP-9 expression in DU 145 and MDA-MB-231 cells were investigated at early and late time points by Western blot analyses in order to determine whether ESE-16 in combination with 4 Gy radiation affects the invasive and migratory signaling properties in cancer cells. BMP-7 expression was significantly lower at 2 h post-radiation in combination-treated DU 145- (0.65 ± 0.11-fold) and MDA-MB-231 cells (0.66 ± 0.05-fold) (Figure 6). A further reduction was evident at the 24-h time period in MDA-MB-231 cells (0.53 ± 0.05-fold) which was also significantly lower than in cells exposed to only 4 Gy radiation over the same time period (1.03 ± 0.05-fold). Thus, the combined therapy prevents the recovery of BMP-7 levels over 24 h when compared to radiation alone. MMP-9 expression remained significantly lower at both 2- and 24-h time periods in DU 145 cells exposed to the combination treatment (0.69 ± 0.13-fold and 0.70 ± 0.10-fold, respectively) (Figure 6). MDA-MB-231 cells exposed to the combination treatment revealed a significant decrease in MMP-9 expression at 2 h (0.59 ± 0.14-fold) with a further decrease at 24 h (0.23 ± 0.14-fold). Furthermore, combination-treated MDA-MB-231 cells showed significantly lower MMP-9 expression at 24 h when compared to the ESE-16 experimental control (0.62 ± 0.12-fold).

### 3.8. ESE-16 Together with Radiation Inhibits HIF-1α Expression in HUVECs

HIF-1α is highly expressed in hypoxic tumor cores and is responsible for promoting the expression of pro-angiogenic proteins which are essential for tumor neovascularization and survival [52,53]. HIF-1α expression was investigated in treated HUVECs by Western blot analyses at early and late time periods. HIF-1α expression decreased significantly when compared to DMSO at 2- and 24-h time periods in HUVECs exposed to ESE-16 alone (0.34 ± 0.22-fold and 0.24 ± 0.12-fold, respectively) and the combination treatment (0.13 ± 0.03-fold and 0.20 ± 0.09-fold, respectively) (Figure 6). ESE-16/4 Gy radiation combined yielded significantly lower HIF-1α expression when compared to 4 Gy radiation alone at both time periods, 0.96 ± 0.05-fold at 2 h and 1.03 ± 0.04-fold at 24 h.

### 3.9. Pre-Osteoclasts Retentained Theirr Differentiation Capacity after Treatment with the Comvination Therepy, and Actin Ring Formation Was Preserved in Mature Osteoclast

TRAP is highly expressed in mature osteoclasts and serves as a histochemical marker for osteoclastogenesis [46]. The effect of the combination treatment on TRAP in RAW 264.7 macrophages either before or after differentiation was investigated to determine the ability of pre-osteoclasts to maintain their differentiation potential, and to determine if the treatment is toxic to mature osteoclasts. RAW 264.7 macrophages exposed to ESE-16 and 4 Gy radiation prior to differentiation showed very slight (but significant) increases in TRAP activity in the 4 Gy radiation- (1.02 ± 1.50 × 10^−3^-fold) and combination (1.03 ± 0.02-fold) treatment conditions (Figure 7A). In contrast, RAW 264.7 macrophages treated after differentiation into mature osteoclasts showed significant decreases in TRAP activity in all treatment conditions: ESE-16 (0.93 ± 0.05-fold), 4 Gy radiation (0.91 ± 0.04-fold) and the combined therapy (0.92 ± 0.07-fold). Mature multinucleated osteoclasts (≥3 nuclei) were visualized with the TRAP stain (red) and hematoxylin (purple) (Figure 7A). No statistically significant changes were documented in osteoclast numbers when treatment was performed on pre-osteoclasts. In contrast, small but significant reductions in osteoclast numbers were recorded in all treatment conditions when treatment was performed on fully-differentiated mature osteoclasts. Fold-decreases of 0.75 ± 0.10, 0.84 ± 0.05, and 0.79 ± 0.03 were observed in the ESE-16-, 4 Gy radiation, and combination treatments, respectively. Thus, mature osteoclasts display increased susceptibility to treatment compared to pre-osteoclasts, but the combination does not appear to be more toxic than the experimental controls.

Actin rings are responsible for isolating bone matrix from the extracellular environment in order for its degradation during bone resorption [54]. Actin ring formation is thus an indirect indicator of optimal resorptive activity. Since statistically significant effects were observed on treated mature osteoclasts in TRAP experiments, the effects thereof were investigated on actin ring formation. Fluorescent microscopy was employed to visualize actin rings (green) and cell nuclei (blue) (Figure 7B). Large multinucleated osteoclasts with intact actin rings were observed in the RANKL+ and DMSO controls. Compared to the DMSO control, the individual therapies, ESE-16, and 4 Gy radiation alone showed fewer actin rings that were smaller in size. These effects were exacerbated in the combination treatment; however, actin rings remained intact, suggesting that the osteoclasts retained their resorptive capability.

## 4. Discussion

MTAs disrupt microtubules dynamics and halt mitosis at metaphase due to the inability of cells to successfully pass the spindle assembly checkpoint (SAC), resulting in the induction of apoptosis [33]. Sulfamoylated estrone derivatives of 2ME2, such as ESE-16, exhibit improved bioavailability and enhanced potency when compared to the parent compound [36]. Previously, a sister compound of ESE-16 namely 2-ethyl-3-O-sulfamoyl-estra-1,3,5(10),15-tetraen-17-ol (ESE-15-ol) demonstrated radiosensitizing properties in lung carcinoma (A549) and MCF-7 cells. Colony formation assays indicated decreased long-term cell survival and regeneration [55]. This proof of concept study aimed to examine the cell signaling involved in the induction of the cell death by a low-dose drug-radiation combination treatment, and thus earlier timepoints post-exposure were studied. Additionally, components of a bone metastatic lesion were screened for their responses to this treatment in order to identify potential aspects worth examining in greater detail in future.

In this study, ESE-16 displayed cytotoxicity at nanomolar concentration in DU 145 and MDA-MB-231- cancer cells, as well as RAW 264.7 macrophages and HUVECs. Response to radiation was enhanced by pre-exposure to the drug. MC3T3-E1 pre-osteoblastic bone cells displayed relative resistance to ESE-16 alone as well as in combination with 4 Gy radiation.

In previously published research, a significant increase in the mitotic and sub-G_1_ population with a corresponding reduction in the G_1_ population was reported on cell cycle analysis of MCF-7 cells pre-sensitized with ESE-15-ol prior to 6 Gy irradiation [55]. In this study, DU-145, MDA-MB-231 cells, and HUVECs exposed to ESE-16 and 4 Gy radiation in combination showed similar results. In DU 145 and MDA-MB-231 cells, the combination treatment was more cytotoxic than radiation alone, and caused greater mitotic inhibition than ESE-16 alone. Microscopy findings supported these data, although MDA-MB-231 cells treated with ESE-16 alone or in combination with 4 Gy radiation yielded more a pronounced effect on the MI than DU 145 cells. Both cell lines revealed a significant increase in metaphase cells when compared to the radiation treatment control. The mitotic block displayed in the combination treated cells, together with the decreased cell density demonstrated, was most likely augmented by the microtubule abrogative properties of ESE-16 and the mitotic catastrophe caused by the radiation [56].

The combined effects of ESE-16 and 4 Gy radiation on cell survival were evident by a significant decrease in DU 145 and MDA-MB-231 cell viability and a significant increase in apoptosis. These effects surpassed those generated in the ESE-16 and radiation controls, suggesting enhanced or augmented potency. These effects may not yet be striking, as early timepoints were investigated in an effort to understand the signaling and the cellular interplay of the two modalities. These results correlate with a previous study investigating the radiosensitizing effects of ESE-15-ol [55]. A significant reduction in cell viability with a concurrent increase in apoptosis was observed in A549 and MCF-7 cells pre-exposed to low-dose ESE-15-ol prior to 6 Gy irradiation. Long-term survival was significantly decreased in the combination treatment as determined by clonogenic studies.

Caspase 3 (CPP32) activation is a landmark event of caspase-dependent apoptosis [57,58]. MDA-MB-231 and HeLa cells exposed to 0.2 µM of ESE-16 yielded a small but significant increase in CPP32 activity after a 24-h-exposure, an effect that was amplified by an increased dose [38,39,59]. In this study, the positive method control (paclitaxel) confirmed that the DU 145 and MDA-MB-231 cell lines used express caspase 3. Combination-treated MDA-MB-231 cells revealed involvement of CPP32 in the cell death signaling which declined from 24 to 48 h. DU 145 cells did not show an increased CPP32 activation in response to treatment at the studied time periods and at the low treatment doses. It is possible that late apoptotic events involving caspase 3 activity have not yet occurred at these time periods or that executioner caspases 6 and/or -8 may alternatively be involved in concluding apoptosis [60]. Temporal activation studies would aim to give clarity to this hypothesis, as well as investigation into caspase-independent death pathways.

ROS, such as O_2_^−^, are chemically reactive oxygen-containing molecules that are generated as a by-product of normal oxygen metabolism and radiation exposure. Since ROS interact with intracellular biomolecules, an imbalance in ROS generating and scavenging systems has the potential to damage DNA, and induce intrinsic apoptotic pathways [61]. Oesophageal carcinoma (SNO), MCF-7, and MDA-MB-231 cells exposed to ESE-16 generated increased O_2_^−^ levels in a temporal response to the drug exposure [38,42]. In this study, O_2_^−^ levels in HUVECs increased significantly when treated with ESE-16 alone or in combination with 4 Gy radiation.

Mn represent residual chromosomal damage resulting from mis- or non-repaired double-strand DNA breaks derived from radiation exposure, and thus serve as an indication of chromosomal radiosensitivity [62,63]. At 2 and 24 h, DU 145 and MDA-MB-231 cells treated with ESE-16 prior to 4 Gy irradiation revealed a significant prevalence of cells containing between 1 and 4 Mn when compared to the ESE-16 treatment control. Moreover, the combined therapy yielded total Mn numbers that were significantly greater than ESE-16 alone but not statistically different from radiation alone. However, higher Mn numbers per cell that were observed in the combination treatment suggest a greater extent of DNA damage than caused by radiation alone. ESE-16 itself does not appear to have genotoxicity. ESE-16 may facilitate Mn formation when combined with radiation due to its effects on the microtubule dynamics which result abrogative cell division processes (including a prophase and metaphase block), and could delay DNA repair by altering DNA damage and repair signaling due to compromised protein shuttling [64].

γ-H2A.X foci form at phosphorylated sites and are directly proportional to the extent of DNA damage [65,66]. 2ME2 in combination with 6 Gy X-radiation increased the formation of γ-H2A.X foci in radioresistant MDA-MB-435R cells, more so than radiation exclusively [31]. In this study, histone H2A.X phosphorylation increased significantly in combination-treated DU 145 and MDA-MB-231 cells when compared to DMSO, an observation absent in the individual treatment controls. Since Kim et al. observed higher γ-H2A.X expression in prostatospheres than in adherent LNCaP cells following exposure to 10 Gy ionizing radiation [61], these results would do well to be repeated in a 3-D spheroid model. Fluorescent γ-H2A.X foci were evident in all treatment conditions with pan-nuclear staining possibly indicating apoptotic bodies or DNA fragments [67].

The osteogenic protein BMP-7 and endopeptidase MMP-9 are upregulated in both breast and prostate cancer bone metastases. They facilitate mesenchymal-epithelial transition, promote invasive and migratory properties of tumor cells to bone sites and support distant tumor survival within the bone microenvironment [12,68]. BMP-7 and MMP-9 expression was inhibited in DU 145 and MDA-MB-231 cells exposed to ESE-16 and 4 Gy radiation. Expression declined further at 24 h in MDA-MB-231 cells, whereas DU 145 cells indicated similar inhibitory effects at both time periods. The combination treatment delayed recovery of BMP-7 expression in MDA-MB-231 cells, as seen in the radiation control. This may be suggestive of impaired signaling between the primary tumor cells to the bone sites, as well as impaired survival of metastatic tumor cells within the bone microenvironment. BMP-7 expression is higher in breast cancer than in prostate cancer [68]. This may account for the ability of DU 145 cells to stabilise BMP-7 levels more rapidly. A trending decline in MMP-9 in MDA-MB-231 cells may limit tumor cell migration and adhesion to bone as well as decrease osteoclast differentiation and activation [12,69]. Future experiments will include quantification of MMP-9 activity via zymology in cellular co-culture protocols.

HIF-1α promotes the adaptive responses of tumor cells to their hypoxic microenvironment through the downstream transcriptional activation of genes regulating tumor survival and progression. These genes express proteins that facilitate cellular oxygen and nutrient transport such as erythropoietin, VEGF-A, and transferrin [70,71,72]. HIF-1α messenger ribonucleic acid (mRNA) levels, protein expression, and relative density were inhibited in 2ME2-treated MDA-MB-435R cells [31]. In this study, ESE-16 and combination-treated HUVECs displayed significant reductions in HIF-1α expression at 2 and 24 h. Furthermore, HIF-1α expression in the combination treatment condition was significantly lower than observed in the radiation treatment control, attributing this suppression solely to the drug effect.

TRAP is an iron-containing enzyme with high expression in mature osteoclasts and essential involvement in bone resorption [73]. Fully-differentiated mature osteoclasts treated with ESE-16 and 4 Gy radiation were more sensitive than pre-osteoclasts treated prior to differentiation. This suggests that the surviving population of treated pre-osteoclasts maintain their ability to differentiate into mature functional osteoclasts. However, treated mature osteoclasts displayed a significant reduction in osteoclast numbers with a concurrent reduction in TRAP activity.

At the site of bone resorption, mature osteoclasts polarise and attach to the bone surface via actin-rich podosomes which form circular actin rings that isolate bone matrix. Actin ring formation in mature osteoclasts is an important indirect indicator of correct polarization and resorptive activity [54]. 2ME2 demonstrated preservation of bone density in ovariectomized rats, suggesting inhibition of bone resorption [74]. Since TRAP experiments rendered mature osteoclasts more sensitive than pre-osteoclasts, this necessitated further investigation on treated mature osteoclast by examining actin ring formation. Decreased osteoclast densities were accompanied by actin rings that were smaller in size. However, actin ring morphology was retained. These observations suggest that bone resorptive activity may be compromised, but not completely lost. Since treatment had less of a cytotoxic effect on the MC3T3-E1 pre-osteoblasts, the combination treatment could perhaps shift the balance of bone remodelling towards bone formation, and in this way could elicit bone-protective effects. This however would need to be studied in a more holistic cellular model using primary osteoblast cultures and bone marrow-derived macrophages in breast and prostate cancer conditioned media. MDA-MB-231 cells do not express RANKL; however, high tumor necrosis factor-α (TNF-α) secretion by these cells stimulates downstream p38 signaling pathways involved in osteoclast differentiation and inflammatory bone loss [75,76,77].

Thus, ESE-16 may radiosensitize cancer cells by pausing the cell cycle at the sensitive G_2_/M phase, increasing ROS production, causing an abrogative autophagic response, and delaying the DNA damage repair by impeding the cytoplasmic-nuclear translocation of DNA damage repair proteins, amongst others [41,44]. A limitation of this deconstructed monolayer cellular model is the lack of the metastatic microenvironment consisting of various intricate cellular interactions. As an example, soluble breast cancer factors are responsible for the transformation of human mammary fibroblasts into cancer-associated fibroblasts which promote tumor growth and have increased invasive and migratory properties [78]. The results of this study provide a foundation for future studies in which the bone microenvironment may be simulated, including co-culture systems and spheroid models. Establishment of osteolytic metastatic breast cancer cell lines, such as those described by Han et al. may be instrumental in analyzing the effect of the combination treatment on the metastatic mechanisms involved in bone lesions [79].

## 5. Conclusions

This study explored the individual responses of the representative components of bone metastasis in which ESE-16 pre-sensitization enhanced the effects of radiation. Overall, the ESE-16/radiation combination treatment induced a G_2_/M phase arrest (metaphase block) in both cancer and endothelial cells along with apoptosis induction and ROS generation. Early and late Mn analyses indicated sustained radiation-induced nuclear damage in cancer cells with phosphorylation of histone H2A.X (γ-H2A.X foci) indicating double-strand DNA breaks. Inhibition of metastatic signaling through the downregulation of BMP-7 and MMP-9 expression may interrupt bone–tumor communication. Additionally, inhibition of angiogenic signaling through the downregulation of the hypoxic responder HIF-1α may reduce tumor growth. Cytotoxic effects were observed in pre-osteoclasts whilst pre-osteoblasts were spared. Preservation of TRAP activity and actin ring formation suggest maintained pre-osteoclastic differentiation capacity and possible retention of osteoclastic resorptive activity This combined treatment strategy shows promise in the treatment of bone metastases by targeting tumor and endothelial cells whilst sparing bone cells. Thus, the use of non-lethal doses of ESE-16 together with a lower dose of radiation may reduce the occurrence of local and systemic adverse effects. Future research will focus on in vitro 3-dimensional co-culture models to imitate the tumor micromilieu in bone metastasis.

## Figures and Tables

**Figure 1 cells-10-02105-f001:**
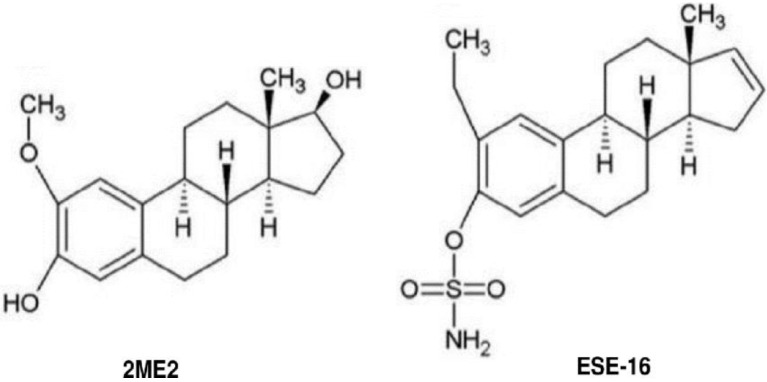
Chemical structures of 2-methoxyestradiol (2ME2) and 2-ethyl-3-*O*-sulfamoyl-estra-1,3,5(10)16-tetraene (ESE-16). The parent compound 2ME2 differs from the analogue ESE-16 with respect to a 2′ ethyl group, 3′ sulfamoylation, and a 17′ alteration.

**Figure 2 cells-10-02105-f002:**
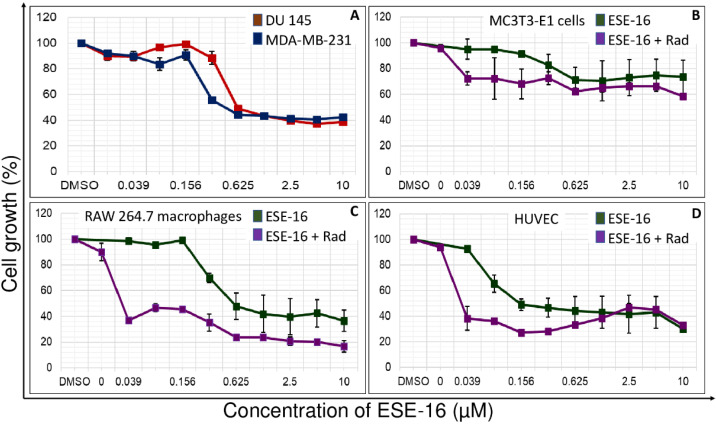
Dose–response curves. (**A**) DU 145 and MDA-MB-231 cells exposed to ESE-16 for 48 h. (**B**) MC3T3-E1 cells, (**C**) RAW 264.7 macrophages, and (**D**) HUVECs exposed to ESE-16 alone as well as ESE-16 in combination with 4 Gy radiation. The 0 µM data point (of the purple line in (**B**–**D**)) represents the effects of 4 Gy radiation alone. Cytotoxic effects were evident in DU 145-, MDA-MB-231-, RAW 264.7 cells, and HUVECs exposed to ESE-16. Sensitivity to 4 Gy radiation increased in the RAW 264.7 and HUVEC cells when they were pre-exposed to low-dose ESE-16. MC3T3-E1 cells were spared. Data points depict the means of a minimum of three biological repeats each with an n of at least 3. Standard deviation (SD) indicated by T-bars.

**Figure 3 cells-10-02105-f003:**
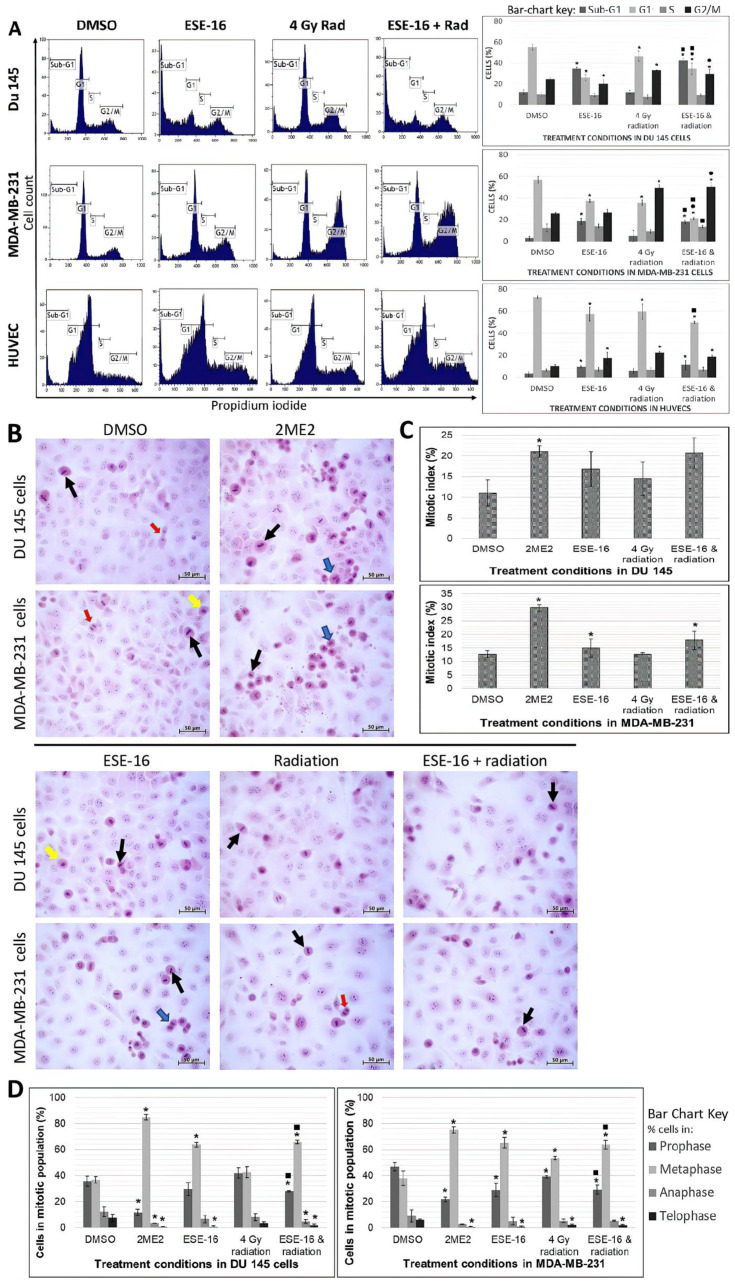
Cell cycle analysis and mitosis studies. (**A**) Flow cytometric histograms of cell cycle progression in DU 145-, MDA-MB-231 cells, and HUVECs exposed to ESE-16 and 4 Gy radiation. Cell count was plotted against PI (FL3). DMSO was used as the vehicle control. Higher sub-G_1_ and G_2_/M populations were accompanied by lower G_1_ populations. (**B**) Hematoxylin and eosin-stained DU 145 and MDA-MB-231 cells exposed to the various treatments (40× magnification, 50 µm scale bar). Metaphase cells (indicated by black arrows) were prevalent in the ESE-16 and combination treatment, more so than in the radiation experimental controls, Yellow arrows = prophase; blue arrows = anaphase; and red arrows = telophase. Higher mitotic indices (**C**) were calculated in the MDA-MB-321 cells treated with ESE-16 and radiation. (**D**) Cells treated with the combination treatment demonstrated more cells in metaphase in both cell lines, similar to the ESE-16-exposed cells. Bar charts represent the means of three biological repeats with SD indicated by T-bars. Statistical significance (*p*-value < 0.05) indicated by * when compared to DMSO, ^●^ when compared to ESE-16, and ^■^ when compared to 4 Gy radiation.

**Figure 4 cells-10-02105-f004:**
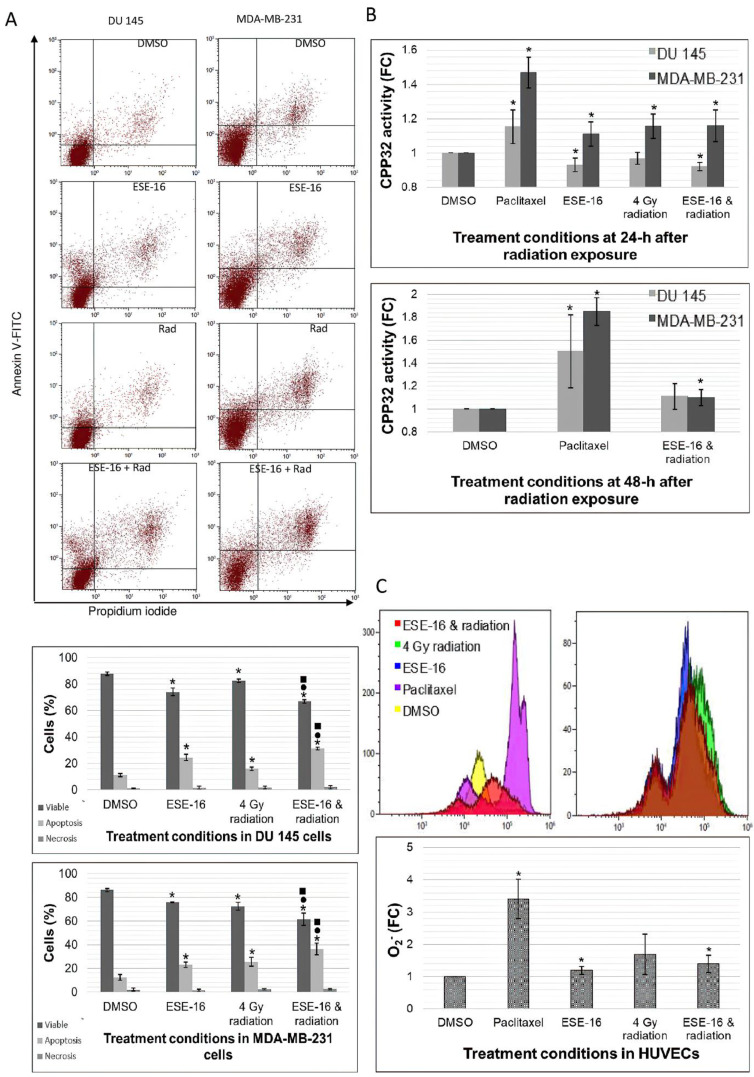
Annexin V-FITC apoptosis detection, caspase 3 activity, and O_2_^−^ detection. (**A**) Flow cytometric dot plots with Annexin V-FITC (FL1) plotted against PI (FL3) of DU 145 and MDA-MB-231 cells exposed to ESE-16 and 4 Gy radiation. DMSO was used as the vehicle control. Apoptotic effects were observed in the ESE-16/4 Gy radiation combination treatment when compared to the treatments individually. (**B**) Graphical representation of CPP32 activity in DU 145 and MDA-MB-231 cells revealed caspase 3-dependent cell death in MDA-MB-231 cells. (**C**) Overlay histograms representing cell count against hydroethidine emission (FL3) in treated HUVECs. Treatment with ESE-16 and 4 Gy radiation increased O_2_^−^ levels. Bar charts represent the means of three biological repeats with SD indicated by T-bars. Statistical significance (*p*-value < 0.05) indicated by * when compared to DMSO, ^●^ when compared to ESE-16, and ^■^ when compared to 4 Gy radiation.

**Figure 5 cells-10-02105-f005:**
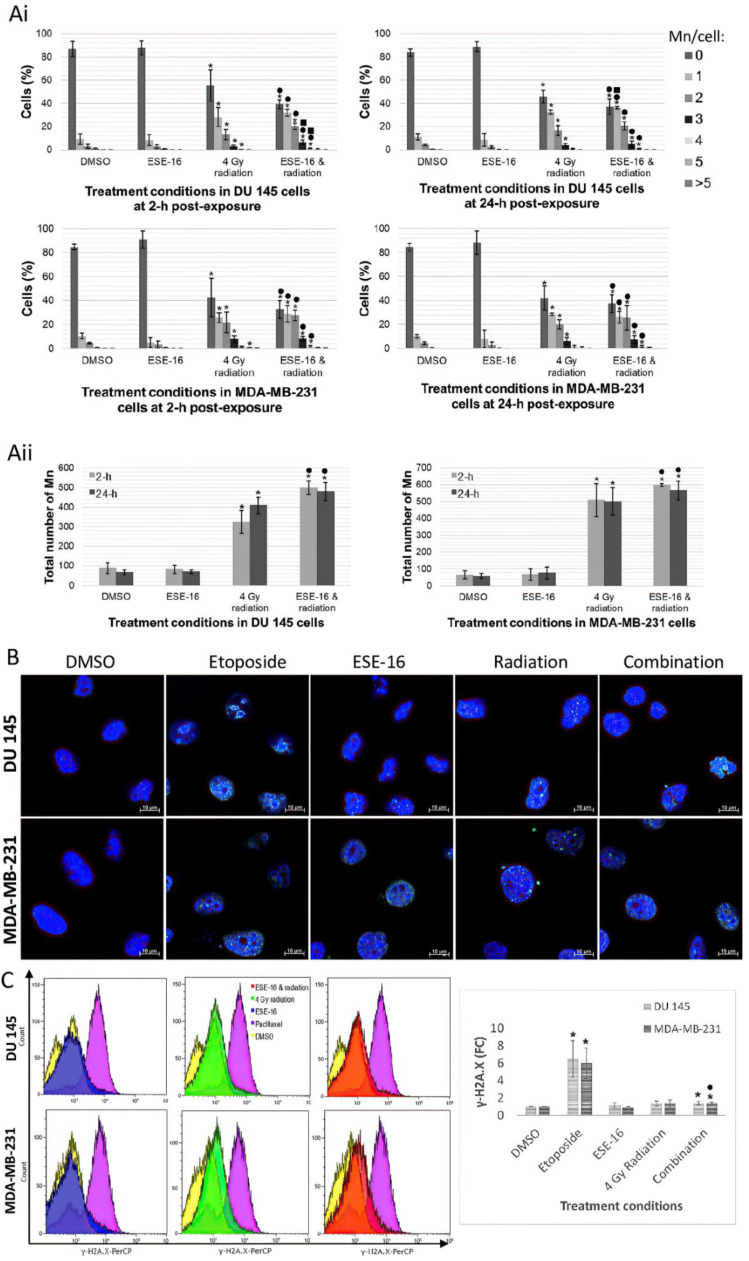
Investigation of DNA damage via Mn and histone H2A.X phosphorylation (γ-H2A.X). (**A**) Mn analyses of DU 145 and MDA-MB-231 cells exposed to ESE-16 and 4 Gy radiation at 2 and 24 h post-treatment. The 4 Gy radiation and combination treatments revealed similar total Mn numbers (**Aii**). However, the combined therapy generated more extensive damage quantified as the percentage of cells with an increased number of Mn per cell (**Ai**). (**B**) Fluorescent micrographs showed increased γ-H2A.X foci (green) within nuclei as well as outside of nuclei (10× magnification, 10 µm scale bar). (**C**) Overlay histograms representing cell count against γ-H2A.X-PerCP (FL4) of DU 145 and MDA-MB-231 cells exposed to ESE-16 and 4 Gy radiation. DMSO was the vehicle control and etoposide served as a positive method control for DNA damage (γ-H2A.X). Treatment with ESE-16 and 4 Gy radiation increased H2A.X phosphorylation. Bar charts represent the means of three biological repeats with SD indicated by T-bars. Statistical significance (*p*-value < 0.05) indicated by * when compared to DMSO, ^●^ when compared to ESE-16, and ^■^ when compared to 4 Gy radiation.

**Figure 6 cells-10-02105-f006:**
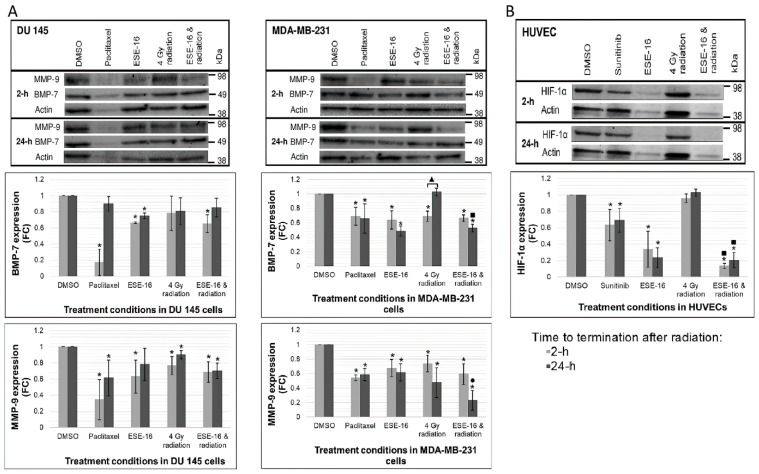
Western blot analyses of metastatic signaling proteins BMP-7 and MMP-9 in cancer cells (**A**), and pro-angiogenic signaling protein HIF-1α in HUVECS (**B**) at 2 and 24 h post-treatment. DMSO was the vehicle control and served as a baseline. The positive method controls were paclitaxel (BMP-7 and MMP-9 inhibition) and sunitinib (HIF-1α inhibition). Separate gels were used for each cell line and each time point and the relevant bands were identified against the protein standard molecular weights (MW). Membranes were cut to quantify the proteins of different MWs. BMP-7 (MW: 49 kDa) and MMP (MW: 92 kDa) expression decreased in DU 145 and MDA-MB-231 cells exposed to ESE-16 and radiation. MDA-MB-231 cells displayed delayed recovery of BMP-7 over 24 h when compared to the radiation control. Prominent reductions in HIF-1α (MW: 93 kDa) expression were evident in HUVECs exposed to ESE-16 alone and the combination treatment. Actin (MW: 42 kDa) served as a housekeeping protein to which protein concentrations were standardized. Bar charts represent mean fold-change (FC) of three biological repeats with SD indicated by T-bars. Statistical significance (*p*-value < 0.05) indicated by * when compared to DMSO, ^●^ when compared to ESE-16, and ^■^ when compared to 4 Gy radiation and ^▲^ comparing 2- and 24 h.

**Figure 7 cells-10-02105-f007:**
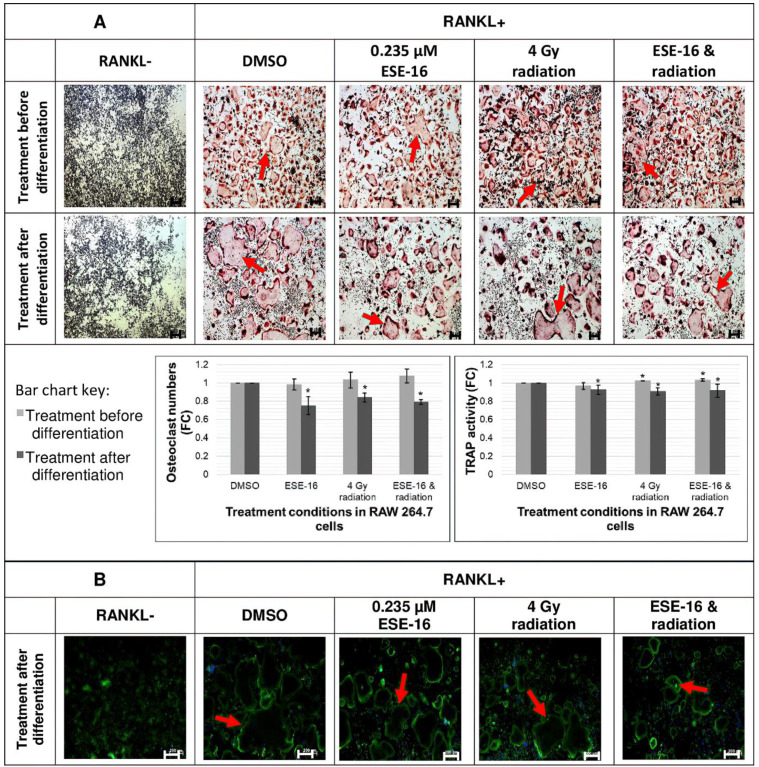
Osteoclast differentiation and activity in RAW 264.7 macrophages exposed to ESE-16 and 4 Gy radiation. (**A**) TRAP activity and staining in RAW 264.7 macrophages treated either before or after differentiation (5× magnification, 200 µm scale bar), with multinucleated osteoclasts indicated by arrows. The effects of ESE-16 and radiation on osteoclast differentiation were more notable in RAW 264.7 macrophages treated after differentiation. (**B**) Actin ring formation in RAW 264.7 macrophages treated after differentiation (5× magnification, 200 µm scale bar). Treated mature osteoclasts showed fewer osteoclasts with intact actin ring structures (indicated by arrows). Bar charts represent the means of three biological repeats with SD indicated by T-bars. Statistical significance (*p*-value < 0.05) indicated by * when compared to DMSO.

**Table 1 cells-10-02105-t001:** GI_50_ values ± SD in DU 145-, MDA-MB-231-, MC3T3-E1-, RAW 264.7 cells, and HUVECs exposed to ESE-16 for 48 h as well as in combination with 4 Gy radiation.

Cell lines	GI_50_ (µM) ± SD	
**Cancer cells**		
	**ESE-16-only dose curve**	
**DU 145**	0.625 ± 0.039	
**MDA-MB-231**	0.469 ± 0.078	
**Bone- and endothelial cells**		
	**ESE-16**	**ESE-16 & 4 Gy radiation**
**MC3T3-E1**	>10	>10
**RAW 264.7**	0.625 ± 0.156	<0.039
**HUVEC**	0.156 ± 0.02	<0.039

## Data Availability

Not applicable.

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
