# Peer review of "Intracellular Signaling Responses Induced by Radiation within an In Vitro Bone Metastasis Model after Pre-Treatment with an Estrone Analogue"

_cells, 2021, doi:10.3390/cells10082105_

Round 1
Reviewer 1 Report
In this study, the authors evaluated the combination therapy of ESE16 and radiation with breast, prostate cancer bone metastasis.
This study showed the effect of the therapy against cancer cells, endothelial cells, and osteoclasts.
However, there are several concerns for the publication in the Cells.
The authors should address all my concerns.
1. The manuscript is too long, and the number of citations exceeds the limit. The long manuscript makes it difficult to understand the novelty and important point of the study. The authors should delete the number of pages. Around 20-25 pages are better. The maximum number of references is around 80.
In addition, introduction and discussion are too long. The authors should cite the reviews approximately, and summarize briefly. For example, LINE43-121 will be able to summarize by citing the review (Maurizi et al., Cancers, 2018; Nakayama et al., Int J Mol Sci., 2021; Gobel et al., Int J Mol Sci., 2021).
2. The authors performed the comparative analysis of the ESE-16 treatment and the ESE-16/Radiation treatment. The lengthly manuscript makes it difficult to understand the novelty.
The authors should rewrite and reconstruct the manuscript logically.
3. Most prostate cancer occurs osteoblastic bone metastasis. The authors should show the osteoblastic phenotype with the treatment.
4. High bone metastatic cell lines are established by previous researches (Kang et al., Cancer Cell, 2003; Eck et al., Breast Cancer Res., 2009; Han et al., Genes Cells, 2020; Park et al., Methods Mol Biol., 2019). The cancer cells in metastatic sites changes the molecular characters and drug sensitivity compared with those parental cells.
Do these therapy have the effect against the high metastatic cell lines?
The authors should discuss the therapy against metastatic cell lines.
5. The citation 64-68 should be deleted.
6. I recommend to recheck the format of references.
7. The resolution of figures is too low to see.
Reviewer 2 Report
In this study, Helena et al. investigated the effects of ESE-16 in sensitising bone metastatic prostate and breast cancer cells and cells of the metastatic niche. ESE-16 treatment prior to radiation was found to induce anti-tumor effects in cancer cells and to reduce neovascularisation while bone cells were not affected by this treatment. The authors suggest the co-treatment of ESE-16 and radiation for patient with bone metastatic disease.
In this study, comprehensive experiments have been conducted to explore the effects of ESE-16 in sensitising prostate and breast cancer cells to radiation. The results are well described and discussed in detail. I have the following suggestions to improve the manuscript:
- The introduction seems to be very long although the major processes during bone metastases are well but to detailed described. The authors might decrease the length of the introduction.
- Figure 2 shows dose-dependent response of DU145 and MDA-MB-231 cells to ESE-16 treatment, but the comparison between effects with and without radiation after ESE-16 treatment are lacking (as shown for the other cell types) and should be included.
- In Figure 3 B, arrows highlight cells in metaphase. Images showing cells in prophase, anaphase and telophase should be added. Who did the analysis and was it blinded to the treatment conditions, cell cycle results etc?
- Western blot showing HIF1-alpha expression in HUVEC cells (Fig. 6) shows variable Actin bands, thus the interpretation of lower HIF-1 alpha in ESE-16 and combinational treatment is improper since Actin levels are reduced as well. Similarly, Actin is reduced after Paclitaxel, thus MMP9/MMP7 reduction can’t be concluded here.
- Figure 6 should be subdivided into subsection (A, B,…).
- The authors might discuss whether the effects after ESE-16 treatment alone, radiation alone or its combination are more likely additive or synergistic based on their results.
- In patients’ treatment, radiation is conducted repeatedly. It would be valuable to investigate long-term or repeatedly effects of ESE-16 +/- radiation in particular for osteoclast differentiation.
- What is known about effects of ESE-16 on benign prostate cells (cytotoxicity)?
Round 2
Reviewer 1 Report
The authors addressed all my concerns.